# A history-dependent approach for accurate initial condition estimation in epidemic models

Dongju Lim[1,2☉], Kyeong Tae Ko[3☉], Hyukpyo Hong[4], Hyojung Lee[3], Boseung Choi[2,5,6], Won Chang[7,8], Sunhwa Choi[9]*, Jae Kyoung Kim[1,2,10]*

1 Department of Mathematical Sciences, KAIST, Daejeon, Republic of Korea, 2 Biomedical Mathematics Group, Pioneer Research Center for Mathematical and Computational Sciences, Institute for Basic Science, Daejeon, Republic of Korea, 3 Department of Statistics, Kyungpook National University, Daegu, Republic of Korea, 4 Department of Mathematics, University of Wisconsin–Madison, Madison, Wisconsin, United States of America, 5 Department of Big Data Science, Korea University, Sejong, Republic of Korea, 6 College of Public Health, The Ohio State University, Ohio, United States of America, 7 Institute for Data Innovation in Science, Seoul National University, Seoul, Republic of Korea, 8 Department of Statistics, Seoul National University, Seoul, Republic of Korea, 9 Innovation Center for Industrial Mathematics, National Institute for Mathematical Sciences, Seongnam, Republic of Korea, 10 Department of Medicine, College of Medicine, Korea University, Seoul, Republic of Korea

☉ These authors contributed equally to this work.
* jaekkim@kaist.ac.kr (J.K.K.), shchoi@nims.re.kr (S.C.)

## Abstract

Mathematical modeling is a powerful tool for understanding and predicting complex dynamical systems, ranging from gene regulatory networks to population-level dynamics. However, model predictions are highly sensitive to initial conditions, which are often unknown. In infectious disease models, for instance, the initial number of exposed individuals (E) at the time the model simulation starts is frequently unknown. This initial condition has often been estimated using an unrealistic, history-independent assumption for simplicity: the chance that an exposed individual becomes infectious is the same regardless of the timing of their exposure (i.e., exposure history). Here, we show that this history-independent method can yield serious bias in the estimation of the initial condition. To address this, we developed a *history-dependent* initial condition estimation method derived from a master equation expressing the time-varying likelihood of becoming infectious during a latent period. Our method consistently outperformed the history-independent method across various scenarios, including those with measurement errors and abrupt shifts in epidemics, for example, due to vaccination. In particular, our method reduced estimation error by 55% compared to the previous method in real-world COVID-19 data from Seoul, Republic of Korea, which includes likely infection dates, allowing us to obtain the true initial condition. This advancement of initial condition estimation enhances the precision of epidemic modeling, ultimately supporting more effective public health policies. We also provide a user-friendly package, Hist-D, to facilitate the use of this history-dependent initial condition estimation method.

**Data availability statement:** The simulated data on daily exposed and infectious people generated in this study is provided as a CSV file 'SimulData_cde.csv' within the Github repository (https://github.com/Mathbiomed/Hist-D), which is publicly available, and a permanent reference to the version of the code used in this study is provided at the Zenodo repository (https://doi.org/10.5281/zenodo.16891923). The real-world confirmed cases and contact tracing data were collected with informed consent and were provided by the Seoul Metro Infectious Disease Research Center. These data are protected and are not available due to data privacy laws. Specific academic requests for access to these data should be directed to the Citizens' Health Bureau, Infectious Disease Control Division, Seoul Metropolitan Government (Tel: 82-02-2133-9480, E-mail: pr77889@seoul.go.kr) or the Institute for Basic Science (E-mail: webmaster@ibs.re.kr).

**Funding:** This work was supported by the National Research Foundation of Korea (NRF) (grant no. RS-202300245056, B.C., D.L., RS-2024-00340139, S.C., NRF-2022R1A5A1033624, H.L., RS- 2022-NR068758, J.K.K., RS-2025-00523567, W.C.), the Institute for Basic Science (grant no. IBS-R029-C3, J.K.K.), the Samsung Science and Technology Foundation (grant no. SSTF-BA1902-01, J.K.K.), a grant of the project The Government-wide R&D to Advance Infectious Disease Prevention and Control (grant no. HG23C1629, B.C., S.C., and H.L.), the New Faculty Startup Fund from Seoul National University (grant no: 326-20240027, W.C.). The funders had no role in study design, data collection and analysis, decision to publish, or preparation of the manuscript. No authors received a salary from any of the funders listed above specifically for this work.

**Competing interests:** The authors have declared that no competing interests exist.

## Author summary

Accurately predicting infectious disease spread requires knowing the initial number of individuals in the exposed compartment at the start of the simulation ($E(t_0)$), but this number is usually unknown. A common method to estimate $E(t_0)$ assumes that the chance of an exposed individual becoming infectious is the same, regardless of when they were exposed. However, this unrealistic assumption can lead to serious errors in the estimation of $E(t_0)$. To solve this problem, we developed a method that considers exposure timing. Our method successfully estimated $E(t_0)$ even with measurement errors or sudden changes in outbreak conditions. In particular, our approach accurately estimated $E(t_0)$ for COVID-19 data from Seoul that includes likely infection dates, which allowed us to obtain the true initial condition. This advancement of initial condition will help improve epidemic predictions and public health strategies. Our method can also be applied to estimate initial conditions in systems where timing or history matters, such as protein maturation or cell degradation pathways. To facilitate the broad adoption of our method, we have also developed and released Hist-D, a user-friendly software package.

## Introduction

Epidemic dynamics have been successfully explained by harnessing mathematical models such as the Susceptible–Exposed–Infectious–Removed (SEIR) model [1–3]. These models predict the future exposed or infectious population over time, allowing predictions of disease spread and the formulation of appropriate public health policies [4–6]. However, these predictions from mathematical models, particularly those based on ordinary differential equations (ODEs), are highly sensitive to the initial condition used, such as the initial number of exposed (E) and infectious (I) individuals. Variations in these initial conditions lead to differences in the simulation of epidemic dynamics [7], ultimately affecting the subsequent estimations of epidemiological parameters such as reproduction number ($\Re$).

Despite the importance of accurate initial conditions for the predictive power of the model, the initial values for some compartments of the model are usually unknown. In particular, the initial condition of the exposed compartment (E) is generally unknown, as determining how many people are actually exposed requires extensive contact tracing, whose complexity increases exponentially with the number of contacts [8]. As a result, previous studies have often subjectively determined the initial condition [9,10]. Some studies minimized this subjectivity by treating initial conditions as free parameters and estimating them [11,12], or using various potential values for the initial conditions and selecting the one whose subsequent simulation best fits the data [13,14]. However, this approach is computationally intensive. An alternative approach estimates the initial condition of E using the known number of daily incidence of becoming infectious [15]. This approach is consistent with the fundamental

assumption of a standard SEIR model—the daily number of new infectious people is the product of (i) the population in the exposed compartment and (ii) the rate of progression to the infectious stage, which is reciprocal to the length of the latent period. Under this assumption, the initial condition of E can be estimated by multiplying the number of daily infectious individuals with the average latent period [15]. However, this method does not account for the different timing of exposure among people in compartment E, instead assuming the same likelihood of transitioning to the infectious stage for all individuals regardless of when they were exposed (i.e., exposure history). Thus, we refer to this method as the History-Independent estimation (Hist-I) throughout this study.

Relaxing the unrealistic history-independent assumption of a standard SEIR model requires two components: (i) a model reflecting the changing likelihood of transitioning to the infectious stage, and (ii) accurate initial conditions for such a model. The first component has been extensively studied through approaches such as the method of stages [16], linear chain tricks [17,18], and delay differential equation (DDE) models [19]. Applying this model, particularly the DDE-based model, enabled more accurate estimation of epidemic parameters by incorporating individual exposure history [19]. However, despite this advantage, a method for accurately estimating their initial conditions—particularly the number of exposed individuals—remains unknown.

Here, we developed a history-dependent method for estimating the initial condition of E, Hist-D, that considers the exposure history. Specifically, we estimated the initial condition of E by finding the solution of the formula expressing the time-varying likelihood of being infectious during a latent period. When applying this approach to simulation data mimicking the latent period of COVID-19, Hist-D outperformed Hist-I under various conditions including scenarios without measurement errors, in the presence of measurement errors, and with abrupt changes in the epidemic phase. Furthermore, when we applied Hist-D to real-world COVID-19 data from Seoul, South Korea, the error in initial condition estimation was reduced by 55% compared to Hist-I. As our approach provides a more accurate estimation of the initial condition of E, it will lead to a more precise understanding of epidemic dynamics, ultimately enabling more effective public health policies. To facilitate the application of Hist-D, we developed a user-friendly package.

## Results

### The history-independent method is inaccurate when the latent period is non-exponential

Mathematical models, even with identical parameter values, can yield different simulation results depending on their initial conditions. Consequently, using different initial conditions to fit the same model to identical data also can yield different estimates of key parameters, such as the transmission rate ($\beta$) or the reproduction number ($\Re$) in the SEIR model (Fig 1a inset). For example, if the initial condition for the exposed population is reduced to 25% of the original value, this leads to a 33.9% relative error in estimating the reproduction number under the parameter condition mimicking the COVID-19 dynamics, highlighting the importance of setting accurate initial conditions (S1a Fig). How the bias in initial conditions evolves in the estimation of the reproduction number is illustrated in S2 Text and S1b Fig.

To calculate this initial condition accurately, it is first necessary to precisely determine the beginning time of the disease (time = 0) and estimate how the epidemic dynamics have changed from that point up to the start of the SEIR model simulation (time = $t_0$) (Fig 1b). For example, to determine the initial condition for the exposed population (E), it is essential to track how E has changed from the beginning of the disease to the start of the simulation. Tracking these daily changes requires knowing how many people became exposed each day ($f_{S \to E}$) and how many transitioned to the infectious stage, leaving the E compartment ($f_{E \to I}$). This information can be derived from exposure history (the date of exposure) and the timing of infectiousness among exposed individuals. However, collecting such data, particularly examining the exposure history of each exposed individual, becomes increasingly challenging as the disease progresses because it requires labor-intensive contact tracing. Consequently, the only available data for determining the initial condition is typically the $f_{E \to I}$ after the data collection starts (Fig 1b) [20], making it challenging to set accurate initial conditions.

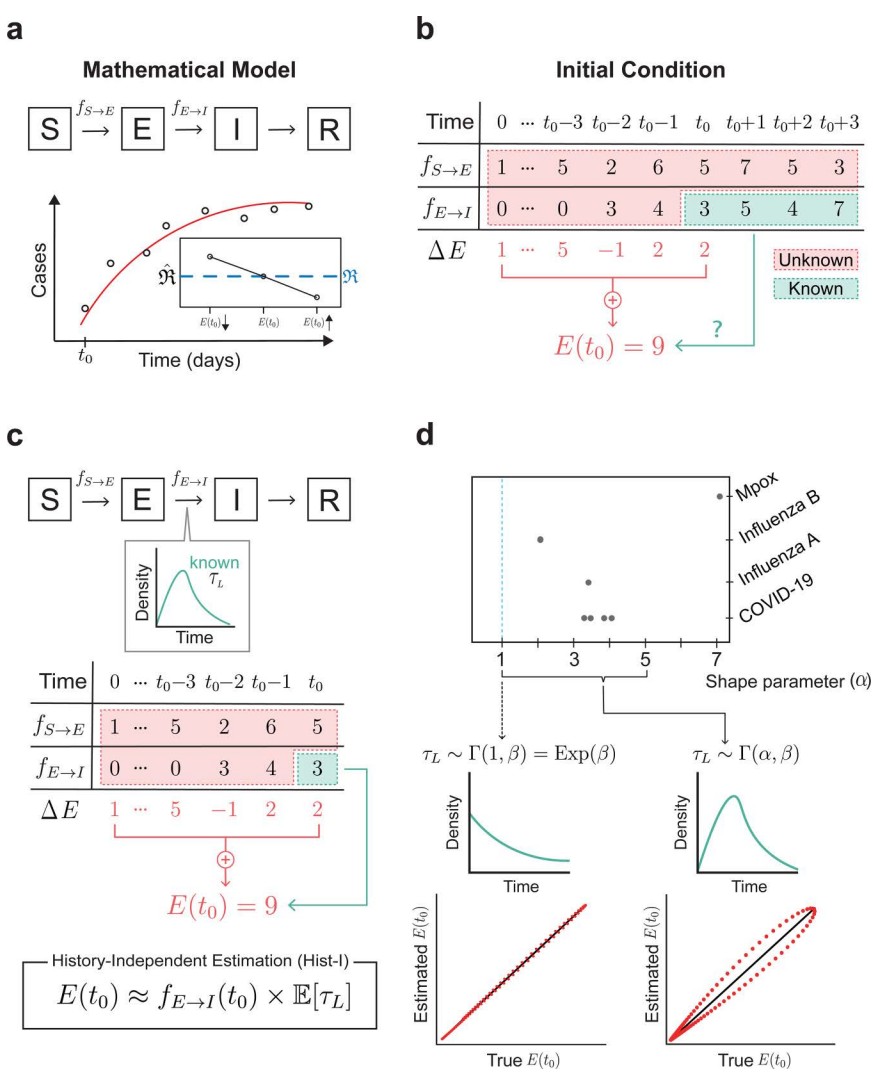

**Fig 1. Estimating initial conditions for the SEIR model.** **(a)** Schematic of the SEIR mathematical model, including the susceptible (S), exposed (E), infectious (I), and removed (R) individuals, which effectively explains epidemic dynamics. Fitting the SEIR model with observed time series (white dots) from $t_0$ enables the estimation of crucial parameters in the epidemic such as reproduction number ($\mathfrak{R}$). This estimation strongly depends on the initial condition at $t_0$ (e.g., $E(t_0)$), the starting point of the model simulation (See Supplementary Information for more details). **(b)** The initial condition of E ($E(t_0)$ in red) can be determined by summing up the daily change of E ($\Delta E$) up to the $t_0$ since the beginning of the disease (0) (red arrow). However, it requires daily incidence of exposure ($f_{S \to E}$) and daily incidence of becoming infectious ($f_{E \to I}$) data before $t_0$, which are often unknown. This highlights the need for a method to estimate the initial condition of E using only the available daily data on infectious individuals from time $t_0$ onward (green arrow). **(c)** To address this limitation, previous studies estimated the initial condition of E ($E(t_0)$ in red) by multiplying $f_{E \to I}$ at $t_0$ and the mean latent period ($\mathbb{E}[\tau_L]$) (green arrow). **(d)** However, while this History-Independent estimation (Hist-I) method provides an accurate estimation (red dots) if the latent period follows the exponential distribution (left), it becomes less reliable for the gamma distribution (right) observed in many infectious diseases, whereby an individual is more likely to transition from exposed to infectious the longer their time since exposure.

To overcome these limitations and set the initial conditions using the available data, the assumption underlying the standard SEIR model can be used (Fig 1c; See S2 Text for more details). Specifically, $f_{E \to I}(t_0)$ is the product of exposed populations ($E(t_0)$) and the rate of becoming infectious ($\kappa$), which is the inverse of the average latent period ($\mathbb{E}[\tau_L] = 1/\kappa$). This assumption naturally leads to the equation $E(t_0) = f_{E \to I}(t_0) \times \mathbb{E}[\tau_L]$: the initial condition of E can be estimated by multiplying the average latent period ($\mathbb{E}[\tau_L]$) by the $f_{E \to I}$ at the initial time point ($f_{E \to I}(t_0)$) [15].

This method, Hist-I, follows the core assumption of the standard SEIR model that does not consider the time when individuals are exposed (i.e., exposure history), assuming everyone experiences the same chance of becoming infectious regardless of their exposure history (Fig 1d). This history-independent (or memoryless) assumption is well-suited for the scenario where the latent period follows an exponential distribution. Conversely, when the latent period follows a non-exponential distribution (e.g., a gamma distribution), the memoryless property is lost, leading to inaccuracies in the Hist-I method (Fig 1d). However, most infectious diseases exhibit a gamma-distributed latent period [21–27] (Fig 1d), meaning that the longer the time since an individual was exposed, the more likely they will transition to becoming infectious (i.e., the history-independent assumption does not hold in reality). This highlights the need for a new method that accounts for this variability in the chance of becoming infectious, depending on the individual's exposure history.

### A framework for estimating initial condition in history-dependent manner

To determine the initial conditions in a history-dependent manner (Fig 2a), we first utilized an equation that represents the relationship between the number of daily exposed individuals each day ($f_{S \to E}(t)$ in Fig 2b) and the number of individuals leaving the compartment E and becoming infectious ($f_{E \to I}(t)$ in Fig 2b), which is given by the data (Fig 2b (i)). This equation uses convolution to express the fact that after being exposed, each individual becomes infectious and leaves compartment E after a latent period ($\tau_L$) following a specific probability distribution ($g_{\tau_L}$; e.g., Gamma) (Fig 2b (i)). In this way, it directly accounts for different exposure histories among exposed individuals.

After discretizing this equation (Fig 2b (ii)), and assuming that individuals were being exposed at a constant rate before time $t_0$ (Fig 2b (iii)), we were able to express the given data ($f_{E \to I}(t_0 + k), k = 0, \ldots, n$) as a linear combination of $E(t_0 - 1)$ and the daily incidence of exposure after time $t_0$ ($f_{S \to E}(t), t = t_0, t_0 + 1, \ldots, t_0 + n$) (See Methods for more details). The coefficient corresponding to $E(t_0 - 1)$ in this linear combination ($Q_{k+1}$ in Fig 2b (iii)) represents the probability that the latent period longer or equal to $k + 1$ days. This expresses that individuals exposed before $t_0 - 1$ must go through a latent period longer or equal to $k + 1$ days to become infectious at time $t_0 + k$. Conversely, the coefficient corresponding to $f_{S \to E}(t_0 + j)$ in the linear combination ($P_{k-j}$ in Fig 2b (iii)) represents the probability that the latent period is $k - j$ days. This reflects that an individual exposed at $t_0 + j$ must go through a latent period of $k - j$ days to become infectious at time $t_0 + k$.

We can write these relationships for all given data ($f_{E \to I}(t_0), \ldots, f_{E \to I}(t_0 + n)$), and by combining them, we can express the equations in the form of a matrix (Fig 2c). By finding the value of $E(t_0 - 1)$ that satisfies this matrix equation, we can estimate the $E(t_0 - 1)$. To achieve this, we created a data loss function that becomes minimal when both sides of the matrix equation are equal (Fig 2d), then sought to minimize this data loss by finding optimal values for unknown parameters ($E(t_0 - 1), f_{S \to E}(t_0), \ldots, f_{S \to E}(t_0 + n)$). However, as the number of parameters to be estimated is $n + 2$, which is one more than the number of data points, $n + 1$, there are infinitely many parameter combinations that satisfy the data loss. To identify a single parameter combination that is close to the true values among these infinitely many combinations, we added a regularization loss term (Fig 2d). This regularization term helps minimize the second derivative of $f_{S \to E}$, ensuring that the estimated $f_{S \to E}$ does not exhibit abrupt changes. Consequently, by finding the combination of $E(t_0 - 1), f_{S \to E}(t_0), \ldots, f_{S \to E}(t_0 + n)$ that minimizes the loss function, which includes both the data loss and the regularization loss, we can estimate $E(t_0 - 1)$. By summing change in the number of exposed people at time $t_0$ ($f_{S \to E}(t_0) - f_{E \to I}(t_0)$), we finally estimated the initial condition of E ($E(t_0)$) (Fig 2d).

### The new history-dependent method outperforms the history-independent method

We evaluated whether our new history-dependent method can provide accurate estimates of the initial condition of E when the latent period follows the gamma distribution unlike Hist-I. To do this, we simulated an SEIR model whose latent period follows the gamma distribution with shape 4.06 and scale 1.35 [24], from $t = 0$ to $t = 200$ (Fig 3a). We then extracted the value of E and the number of people transitioning from E to I ($f_{E \to I}$) at each time point $t$ (see Methods for more details).

PLOS Computational Biology

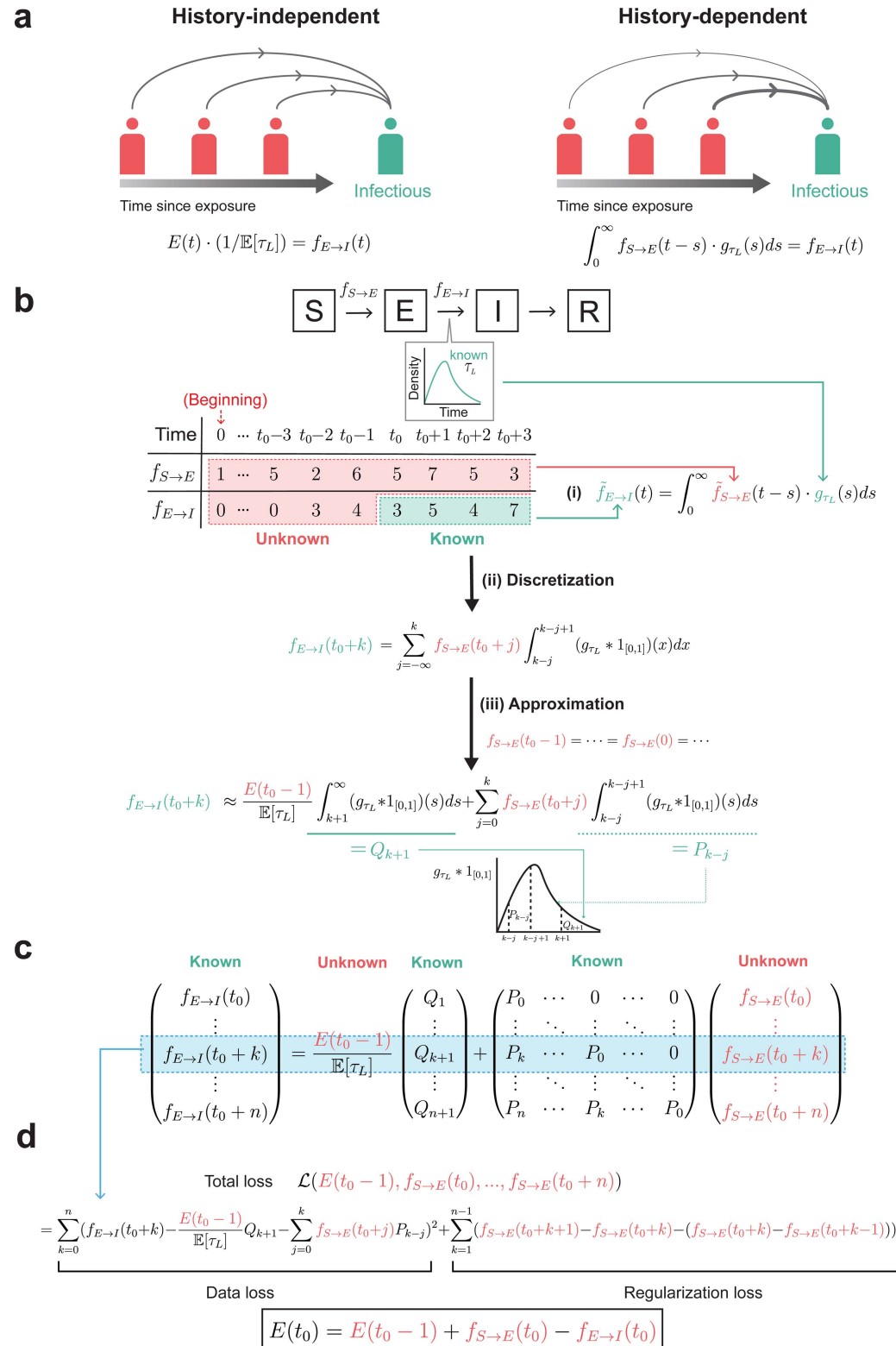

**Fig 2. Schematic figure for deriving the loss function to estimate the initial condition. (a)** To address the limitation of the history-independent method (left), we developed a novel history-dependent method (right). **(b)** (i) We established the connection between the known data, $f_{E\to I}$, and the

unknown $f_{S \to E}$ by treating the $f_{E \to I}$ as a convolutional output of $f_{S \to E}$ and the probability density function of the latent period, $g_{\tau_L}$. (ii) By discretizing this relationship and (iii) assuming $f_{S \to E}$ remains consistent before $t_0$, we can express the known $f_{E \to I}$ as a linear combination of unknown $E(t_0)$ and unknown $f_{S \to E}$ with known coefficients $P_{k-j}$ and $Q_{k+1}$. $P_{k-j}$ represents the probability of an individual having a latent period of exactly $k-j$ days, while $Q_{k+1}$ represents the probability of the latent period being longer or equal to $k+1$ days. $P_{k-j}$ and $Q_{k+1}$ can be obtained by integrating the convolution of $g_{\tau_L}$ and $1_{[0,1]}$, where $1_{[0,1]}$ represents the characteristic function supported on [0,1] (See Methods for more details). **(c)** Extending the linear combination expression to the whole data (i.e., $f_{E \to I}(t)$ for $t = t_0 + 1, \dots$), we can construct a matrix that describes the relationship between known data and unknown parameters. **(d)** We utilized this matrix equation that $E(t_0 - 1)$ must satisfy to establish the data loss function, then sought to minimize this data loss by finding optimal values for unknown parameters, including $E(t_0 - 1)$. However, as the number of unknown parameters ($n + 2$) exceeds the number of equations ($n + 1$), the parameters cannot be determined solely from the data loss. This leads us to incorporate the regularization loss for the $f_{S \to E}$ parameters, which aims to smooth the $f_{S \to E}$ parameters by minimizing their second order derivatives. Consequently, by finding the parameters that minimizes the total loss function ($\mathcal{L}$), which includes both the data loss and the regularization loss, we can estimate $E(t_0 - 1)$. By summing up the difference between daily incidence of exposure ($f_{S \to E}(t_0)$) and daily incidence of becoming infectious at $t_0$ ($f_{E \to I}(t_0)$), we finally get the initial condition of **E.**

With this data, we estimated the initial condition $E(t_0)$ from the given $f_{E \to I}$ data using the history-independent method (Hist-I) and the history-dependent method (Hist-D). First, the Hist-I method estimates the $E(t_0)$ by multiplying the mean latent period (i.e., $4.06 \times 1.35 = 5.48$) by the value of $f_{E \to I}$ at $t_0$ (Fig 3b). For example, we estimated $E(80)$, $E(81)$, $E(82)$, $\dots$, $E(190)$ by multiplying 5.48 by the respective values of $f_{E \to I}$ at $t = 80$, 81, 82, ..., 190. The second method, History-Dependent estimation (Hist-D), estimated the $E(t_0)$ value that minimized the loss function (Fig 2d) with the data of $f_{E \to I}$ for 2 × mean latent periods ≈ 10 days after $t_0$ (Fig 3b). For example, when estimating $E(80)$, we used $f_{E \to I}$ data from $t = 80$ to $t = 90$, and for estimating $E(190)$, we used $f_{E \to I}$ data from $t = 190$ to $t = 200$. Note that while 2 × mean latent periods are used in this study, the length of data can be adjusted by users. Both Hist-I and Hist-D assume that $f_{E \to I}$ data only exists after the time point $t_0$, which is the start of the SEIR model simulation.

Using these methods (Hist-I and Hist-D), we estimated the $E(t_0)$ for $t_0 = 80$, 81, $\dots$, 190 and compared them with their true values (Fig 3c). As a result, Hist-D was much more accurate than Hist-I (Fig 3d), particularly reducing the root mean squared error (RMSE) and mean absolute percentage error (MAPE) by 86% and 85%, respectively (Fig 3e). Similar improvements were also observed during the earlier phase ($t_0 = 10$, $\dots$, 80) of epidemic growth (S3 Fig). This superiority of Hist-D persisted under various parameter conditions (See S1 Table) and even after modifications were made to the Hist-I method by summing up the future daily incidence of becoming infectious, as done in a previous study [28] (see S3 Text and S2 Fig). Consequently, we focused our analysis on the Hist-I method rather than its modified version.

Hist-D demonstrated superior accuracy compared to the Hist-I method under ideal conditions without measurement errors. However, real-world situations differ from simulations, as measurement errors are always present. To simulate a scenario with observation errors, we introduced the multiplicative noise to the given data ($f_{E \to I}$) and used this data to estimate $E(t_0)$ with Hist-I and Hist-D (Fig 3f). When the noise level was 0.1 ($\sigma = 0.1$ in Fig 3f), due to the effect of the multiplicative noise, the error increased as $E(t_0)$ grew larger for both methods (Fig 3g, top). However, Hist-D still maintained smaller errors compared to Hist-I (Fig 3g, top). Furthermore, Hist-D demonstrated greater resilience to increasing noise levels compared to Hist-I, exhibiting a smaller error amplification as the noise intensified from 0 to 0.3 (Fig 3g, bottom). This higher accuracy persisted across all tested noise levels, ranging from 0 to 0.3 in 0.1 intervals (Fig 3h). These results show that Hist-D is recommended for real-world applications with measurement errors in observed data.

Beyond the measurement error, real-world epidemics present additional complexities such as sudden changes in the epidemic phase due to social distancing, vaccination, or large-scale outbreaks of COVID-19. To reflect these changes in the simulation, we regenerated simulation data by changing the transmission rate, $\beta$, at a specific point (i.e., when E reaches its peak) and used this new simulation data to compare the accuracy of Hist-I and Hist-D (Fig 3i). When we doubled the $\beta$ and investigated the errors for the 20-time points before and after the changing point, both Hist-I and Hist-D showed increased errors around the point of the second peak (time = 150 − 160 in Fig 3i; the right most of the graph in Fig 3j). However, the Hist-D method produced smaller errors than the Hist-I method (Fig 3j). This superior performance of Hist-D persisted across various $\beta$ changes (1/3, 1/2, 2, and 3-fold), consistently achieving smaller RMSE and MAPE

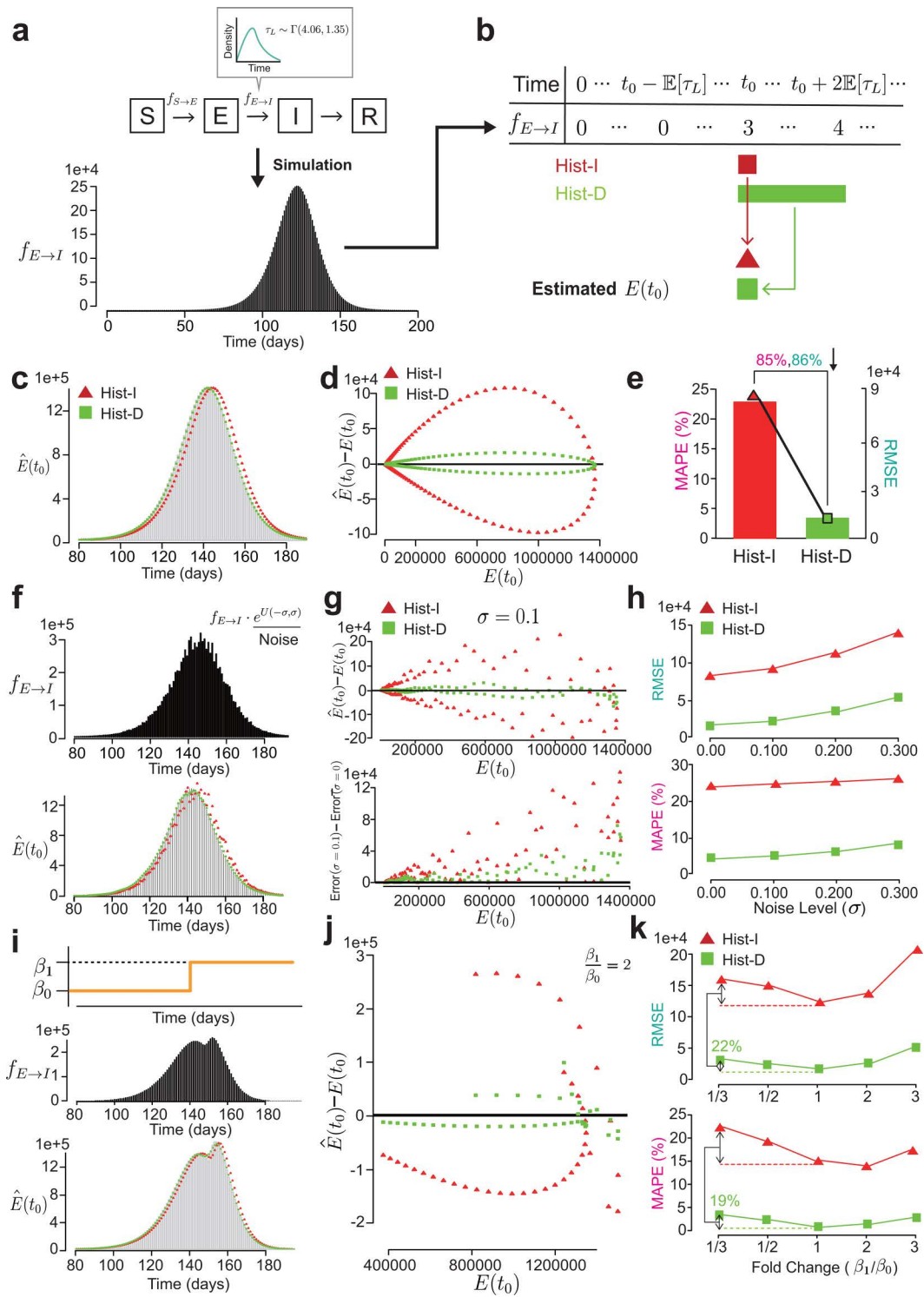

**Fig 3. Hist-D outperforms Hist-I, regardless of the phase transition of epidemic dynamics and noise. (a)** The trajectory of E and the daily incidence of becoming infectious ($f_{E \to I}$) were simulated through the SEIR model whose latent period follows the gamma distribution with shape 4.06 and scale 1.35 (See Methods for more details). **(b)** Simulated $f_{E \to I}$ was then utilized to estimate the $E(t_0)$ and compare History-Independent estimation (Hist-I) and History-Dependent estimation (Hist-D). Hist-I utilizes data from only single day, $t_0$, while Hist-D uses data from $2 \times \mathbb{E}[\tau_L]$ consecutive days

after the $t_0$, where $\mathbb{E}[\tau_L]$ is a mean latent period. **(c)** The graph comparing the true $E(t_0)$ (light gray-colored bars) and the estimated $E(t_0)$ ($\hat{E}(t_0)$). **(d)** The scatter plot displaying the error ($\hat{E}(t_0) - E(t_0)$) across different levels of true $E(t_0)$. Estimation from Hist-D (green squares) has a much lower error compared to Hist-I (red triangles). **(e)** The graph showing the root mean squared error (RMSE) (bars) and the mean absolute percentage error (MAPE) (line) of Hist-I and Hist-D. When Hist-D was utilized, RMSE and MAPE was reduced by 86% and 85%, respectively, compared to Hist-I. **(f)** To better reflect the real-world situation with observation noise in given data, we applied multiplicative noise ($e^{U(-\sigma,\sigma)}$), where $U(-\sigma,\sigma)$ is the uniform distribution on $(-\sigma,\sigma)$, to the simulated $f_{E\to I}$ data used in (c-e) and compared the accuracy of Hist-I and Hist-D. **(g)** The scatter plots displaying the estimation error at the noise level $\sigma = 0.1$. The error of both Hist-I and Hist-D increased proportionally to the level of true $E(t_0)$, and this was specifically manifested in Hist-I (top). In addition, compared to the zero-noise level case (i.e., the case in (c-e)), the error increment of Hist-D was lower than that of Hist-I (bottom). **(h)** The graph showing the RMSE (bars) and MAPE (line) of Hist-I and Hist-D across the different noise levels ($\sigma = 0$, $0.1$, $0.2$, $0.3$). Hist-D achieved a lower RMSE and MAPE than Hist-I across all noise levels. **(i)** To assume the transition of epidemic dynamics, we abruptly changed the transmission rate, $\beta$, from $\beta_0$ to $\beta_1$ at a single point (top), and simulated $f_{E\to I}$ data (middle), which were then used to investigate the accuracy of Hist-I and Hist-D. **(j)** The scatter plot showing the error of Hist-I and Hist-D when the transmission rate has been doubled. Hist-D outperformed Hist-I. **(k)** The graph showing the RMSE (bars) and MAPE (line) of Hist-I and Hist-D across the different fold change ($\beta_1 / \beta_0$ = 1/3, 1/2, 1, 2, 3). Hist-D consistently outperformed Hist-I across all fold changes. In particular, when $\beta$ was reduced to 1/3, the absolute increase in RMSE and MAPE for Hist-D was 22% and 19% that of Hist-I, respectively, demonstrating the robustness of Hist-D to sudden changes in $\beta$.

compared to Hist-I (Fig 3k). In particular, when $\beta$ was reduced to 1/3, the absolute increase in RMSE and MAPE for Hist-D was less than half that of Hist-I (Fig 3k), demonstrating the robustness of Hist-D to sudden changes in $\beta$. Taken together, these results highlight the promising potential of Hist-D for estimating $E(t_0)$ in dynamic, real-world scenarios.

### Hist-D outperforms Hist-I for real-world COVID-19 data

The results from the simulation data demonstrated the strong potential of Hist-D for accurately estimating the initial condition of E in real-world scenarios. To test this, we applied Hist-I and Hist-D to COVID-19 data from Seoul, Republic of Korea, spanning August 13th to November 25th, 2020. This data included the contact dates and symptom onset for people in Seoul, allowing us to empirically derive the number of people moving from S to E ($f_{S\to E}$) and from E to I ($f_{E\to I}$), as well as the distribution of the incubation period (i.e., the time between contact and symptom onset date) (See Methods for more details) (Fig 4a). With this information, we calculated the daily change in E ($f_{S\to E} - f_{E\to I}$) and accumulated these changes starting from the date of the first recorded case of international transmission in Korea, to compute the daily $E(t_0)$ for 2020. Then, this real $E(t_0)$ was compared with $E(t_0)$ estimated by applying Hist-I and Hist-D to the $f_{E\to I}$ data and the empirical distribution of incubation period (Fig 4b). In particular, Hist-D utilized 8 days of $f_{E\to I}$ data, approximately twice the mean incubation period, as in the case of the simulation study (Fig 3b). For the last few days, when 8 days of $f_{E\to I}$ data were unavailable, Hist-D utilized $f_{E\to I}$ data from $t_0$ to the last available date.

While both methods effectively captured the long-term trend of $E(t_0)$, Hist-D exhibited less fluctuation compared to Hist-I (Fig 4b). Notably, Hist-D provided more accurate estimates than Hist-I during abrupt changes in $E(t_0)$ such as near Oct 31 (Fig 4b), consistent with Fig 3k. As a result, Hist-D consistently demonstrated the higher accuracy than Hist-I (Fig 4c, 4d), whose error increased as the magnitude of $E(t_0)$ grew (Fig 4d). In particular, Hist-D reduced the RMSE by 55% compared to Hist-I (Fig 4e). Similarly, it decreased the MAPE by 55% (Fig 4e). These results indicate that in highly volatile real-world scenarios, Hist-D provides more accurate and reliable estimates of initial conditions than the Hist-I method.

Despite the promising results, Hist-D did not achieve perfect estimations. Therefore, we checked whether the true values fell within the 95% credible interval when using the Hist-D (Fig 4f; see Methods for more details). As a result, 91.3% of the true values were included within the credible interval for the Hist-D method (Fig 4f). Taken together, Hist-D demonstrates robust capabilities in precisely determining the initial condition of E, which is likely to result in a more accurate estimation of epidemic dynamics.

### Discussion

While accurate initial conditions are crucial for the SEIR model, the initial condition value of the exposed population (E) is often unknown. Thus, the initial condition of E has often been estimated with the Hist-I method. However, Hist-I does not

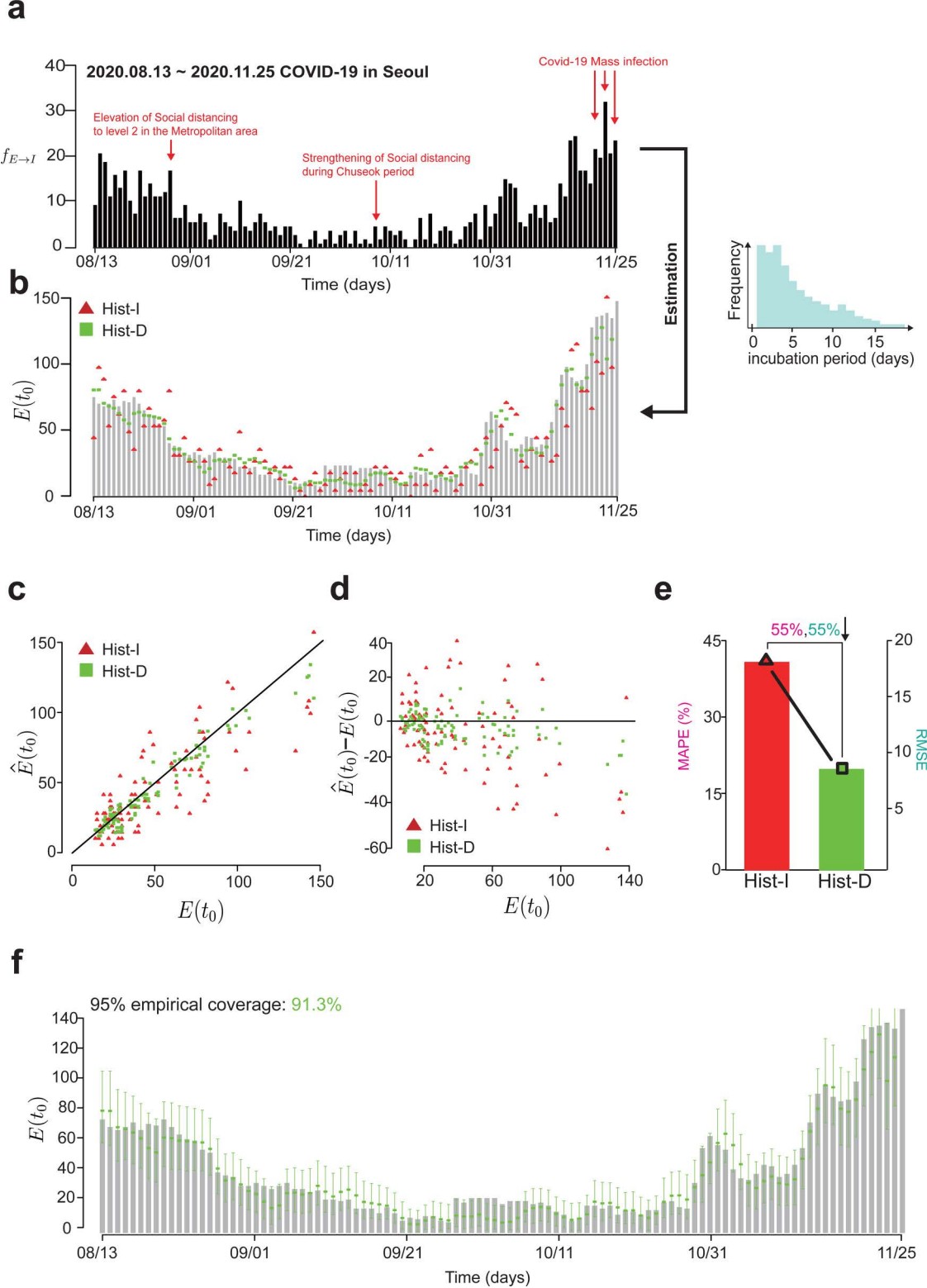

**Fig 4. Hist-D provide more accurate estimates of the initial condition of E compare to Hist-I for real COVID-19 data in Seoul, Republic of Korea. (a)** We compared Hist-I and Hist-D to estimate the initial conditions of E for COVID-19 data in Seoul, Republic of Korea, from August 13 to

November 25, 2020. From this data, $f_{E \to I}$ data and the distribution of the incubation period (light blue histogram) were extracted (see Methods for more details) and then used to estimate the initial condition of E with Hist-I and Hist-D. **(b)** The graph comparing the true $E(t_0)$ (light gray bars) and estimated $E(t_0)$ (Hist-I: red triangles, Hist-D: green squares). While both methods capture the long-term trend, Hist-I exhibits more pronounced fluctuations. **(c)** The scatter plot comparing the true $E(t_0)$ and estimated $E(t_0)$ ($\hat{E}(t_0)$). Estimation from Hist-D is closer to the perfect estimation (i.e., the black cross line, where $E(t_0) = \hat{E}(t_0)$) than Hist-I. **(d)** The scatter plot displaying the error ($\hat{E}(t_0) - E(t_0)$) across different levels of true $E(t_0)$. The error of Hist-I increased proportionally to the $E(t_0)$, while such a pattern was not manifested in Hist-D. **(e)** The graph showing the RMSE (bars) and the mean absolute percentage error (MAPE) (line) of Hist-I and Hist-D. Hist-D achieved 55% lower RMSE (8.44) and 55% lower MAPEs (18.9%) compared to Hist-I (RMSE: 18.76, MAPE: 42.2%), respectively, demonstrating the superior performance of Hist-D, in real-world epidemic data. (f) 95% Credible interval and empirical coverage of our estimated values. The upper and lower horizontal lines of each box represent the upper and lower bounds of the credible interval, corresponding to the 97.5% and 2.5% quantiles, respectively. 91.3% of true values were included in the 95% credible interval of Hist-D.

consider the timing of exposure of the individuals in the exposed compartment (i.e., exposure history). As a result, this method yields biased estimation (Fig 1d). To resolve this problem, in this study, we developed a new history-dependent method, Hist-D (Figs 2 and 3a-3b). For the simulated data, Hist-D estimated the initial condition of E much more accurately than Hist-I (Fig 3c-3e), even with measurement errors (Fig 3f-3h) or sudden changes in epidemic phases (Fig 3i-3k). Importantly, Hist-D successfully estimated the initial condition of E in real-world COVID-19 data from Seoul, Korea, reducing estimation error by 55% compared to Hist-I (Fig 4). These findings demonstrate that Hist-D can more accurately estimate the unknown initial conditions in the SEIR model using relatively accessible data.

Although this study focused on the SEIR model, Hist-D can be applied to any compartmental model where the transition time between two compartments is known and inflow data for the downstream compartment is available. Thus, Hist-D can be used when the daily incidence of becoming infectious and the latent period distribution are known (Fig 3), or when the daily incidence of symptom onset and the incubation period distribution are known (Fig 4). This flexibility allows Hist-D to be applied to other infectious disease models, such as SEIR-Vaccinated (SEIRV) [29,30], SEI-Quarantined-R (SEIQR) [31,32], or SE-Presymptomatic-IR (SEPIR) [33,34]. For more complex models [16] with additional substages in the exposed (E) or infectious (I) compartments, the same approach used in Hist-D can be adapted by modifying the left and right sides of the equations derived in this study (see Fig 2b and the 'Derivation of loss function' section of the Method section for more details). These modifications can also be easily implemented in our Hist-D package by altering a single function (see [7] of the S1 Text for further guidance). Therefore, Hist-D offers a flexible framework that can be readily extended, and applying it to a broader class of infectious disease models represents a promising direction for future research.

The epidemic dynamics, in particular, the transition from exposure to infectiousness, is inherently history-dependent (i.e., its likelihood varies over time since the exposure). However, this has been overlooked in previous studies, which employed a simple ODE model that assumes a constant chance of becoming infectious. While this history-independent representation simplifies the inference of crucial epidemiological parameters such as reproduction number, our previous work revealed that it introduces significant bias [19]. Thus, we address this bias by utilizing a model that describes the history-dependent dynamics [19]. Nonetheless, the advantage of using history-dependent models relies heavily on accurate initial conditions (S4 Text and S4 Fig), as these values significantly affect the model predictions. Previous methods for determining initial conditions were based on a history-independent assumption, misaligning with the dynamics in history-dependent models and resulting in a considerable bias in initial conditions (Fig 3) and subsequent estimation of the reproduction number (S4 Fig). We addressed this here by developing a history-dependent method for estimating the initial condition (S4 Fig). This, combined with history-dependent models, provides the first framework that completely describes the history-dependent dynamics of infectious disease.

Hist-D employs a master equation (Fig 2b (i)), which represents the daily infectious population as a convolution of daily exposed individuals and the latent period distribution. In this study, we modified this master equation to derive the total loss function (Fig 2b-2d). In contrast, previous studies have applied this master equation without direct modification

[35–37]. For example, Abbott et al. utilized a similar master equation to develop an algorithm that can estimate the sometimes-unknown daily infectious population from typically available daily confirmed cases [37]. This suggests the potential to extend the applicability of our approach by combining with the approach by Abbott et al. Specifically, our framework currently requires daily incidence of becoming infectious data, which is sometimes unknown. In such cases, we can estimate the daily infectious cases from the daily confirmed cases, which is typically easier to obtain, by using the approach of Abbott et al.

Beyond infectious disease studies, other biological systems have also been studied using mathematical models incorporating delay [38–46]. These models simplify complex biological processes involving many intermediate stages by representing them as a single pathway with a time delay. For example, the complex maturation process of proteins has been replaced with one single protein production process with time delay [41,43] and the complex degradation pathway of damaged cells has been replaced with a single degradation process with delay [46]. This approach is similar to the SEIR model used in this study, where the detailed process from exposure to infectiousness has been simplified to a single process with delay (i.e., latent period). Considering this, Hist-D can be generalized to the other models incorporating the delay. For instance, when modeling the level of immature and mature proteins, Hist-D could estimate the initial condition of the immature proteins, which is often difficult to measure experimentally, by using the known data of mature proteins.

Despite the novelty of Hist-D, several limitations should be noted. First, our methods are derived from ODE-based infectious disease models, though stochastic compartmental models and network-based models are also used to better capture transmission uncertainty and detailed processes of disease progression, respectively [47,48]. Whether Hist-D can be extended to these models remains an open question, and exploring this would be a promising direction for future research. In addition, Hist-D assumes a constant daily incidence of exposure before the initial time point (See equation (8) in the Methods section). Relaxing this assumption to accommodate scenarios such as super spreading events [49] or exponential growth (or decay) represents another important direction for future work.

Another limitation is that our method has been primarily validated using COVID-19 data. In addition to real data from Seoul, Korea, we used simulation data that mimics the latent period of COVID-19 with a Gamma distribution. However, the latent periods of other infectious diseases may not follow the Gamma distribution. Nonetheless, even in such cases, Hist-D can be readily adapted by simply adjusting the latent period distribution, as our derivation of the loss function does not depend on any specific distributional assumption. Therefore, we hypothesize that Hist-D can still estimate the initial condition of E with reasonably high accuracy across various latent period distributions.

Lastly, the credible intervals for Hist-D were relatively wide, indicating a high degree of uncertainty in the estimates. As such uncertainty can hinder the precise determination of initial conditions, future work should focus on reducing this uncertainty. Additionally, the empirical coverage of the 95% credible intervals was below the expected level (i.e., 95%). This discrepancy may arise from a mismatch between the real-world data generation process and the model assumptions, such as that constant exposure occurs before the initial point (equation (8)), underlying the Bayesian approach. Reducing this model deficiency through advanced statistical techniques [50] could improve empirical coverage and enhance the reliability of the estimates.

## Method

### Derivation of the total loss function used in Hist-D

We established the total loss function to estimate the initial conditions of exposed individuals. This loss function started from the master equation that characterizes the history-dependent rate of becoming infectious by incorporating a non-exponentially distributed latent period. While this can be modeled through the method of stages [16], which introduces multiple substages in the exposed compartment, it requires specifying the number of substages by fitting an Erlang distribution to the empirical distribution of the latent period, which may increase the computational cost of the optimization process. More importantly, the method of stages constrains the latent period to an Erlang distribution. To avoid this, we

adopted an alternative approach, following a previous study [19], which allowed us to incorporate arbitrary latent period distribution. In particular, the instantaneous rate of individuals becoming infectious is a convolution of the instantaneous rate of individuals exposed and the probability density function of the latent period, as follows (Fig 2b (i)):

$$\tilde{f}_{E \to I}(t) = \int_0^\infty \tilde{f}_{S \to E}(t - s) g_{\tau_L}(s) \, ds \tag{1}$$

where $\tilde{f}_{E \to I}(t)$ represents the instantaneous rate of the number of individuals becoming infectious at time $t$, $\tilde{f}_{S \to E}(t)$ denotes the instantaneous rate of the number of individuals exposed at time $t$, and $g_{\tau_L}(s)$ is the probability density function of the latent period. Here, we modified this equation to explicitly account for the effect of the initial condition of E on the daily infectious individuals. We first integrated the equation (1) to express the number of daily infectious individuals ($f_{E \to I}(t_0 + k)$).

$$f_{E \to I}(t_0 + k) = \int_{t_0+k-1}^{t_0+k} \tilde{f}_{E \to I}(t) \, dt = \int_{t_0+k-1}^{t_0+k} \int_0^\infty \tilde{f}_{S \to E}(t - s) g_{\tau_L}(s) \, ds dt \tag{2}$$

Then, we discretized the marginal number of individuals exposed in terms of the number of daily incidence of exposure, as follows (Fig 2b (ii)):

$$\tilde{f}_{S \to E}(t) = f_{S \to E}(t_0 + j) \ \ if \ t_0 + j - 1 < t \le t_0 + j \tag{3}$$

This can be rewritten as follows:

$$\tilde{f}_{S \to E}(t) = \sum_{j=-\infty}^{\infty} f_{S \to E}(t_0 + j) 1_{[t_0+j-1, \, t_0+j]}(t) \tag{4}$$

where $1_{[t_0+j-1, t_0+j]}$ denotes the characteristic function supported on the interval $[t_0 + j - 1, \ t_0 + j]$. By plugging in equation (4) to the equation (2), we derived the new equation:

$$f_{E \to I}(t_0 + k) = \int_{t_0+k-1}^{t_0+k} \int_0^\infty \sum_{j=-\infty}^{\infty} f_{S \to E}(t_0 + j) 1_{[t_0+j-1, \, t_0+j]}(t - s) g_{\tau_L}(s) \, ds dt$$

$$= \sum_{j=-\infty}^{\infty} f_{S \to E}(t_0 + j) \int_{t_0+k-1}^{t_0+k} \int_0^\infty 1_{[t_0+j-1, \, t_0+j]}(t - s) g_{\tau_L}(s) \, ds dt \tag{5}$$

By changing the variable from $t$ to $x$ by $t = t_0 + j - 1 + x$, we obtain

$$f_{E \to I}(t_0 + k) = \sum_{j=-\infty}^{\infty} f_{S \to E}(t_0 + j) \int_{k-j}^{k-j+1} \int_0^\infty 1_{[t_0+j-1, \, t_0+j]}(t_0 + j - 1 + x - s) g_{\tau_L}(s) \, ds dx$$

$$= \sum_{j=-\infty}^{\infty} f_{S \to E}(t_0 + j) \int_{k-j}^{k-j+1} \int_0^\infty 1_{[0, \, 1]}(x - s) g_{\tau_L}(s) \, ds dx \tag{6}$$

$$= \sum_{j=-\infty}^{\infty} f_{S \to E}(t_0 + j) \int_{k-j}^{k-j+1} \left( g_{\tau_L} * 1_{[0,1]} \right)(x) \, dx$$

where * symbol denotes the convolution. Considering that $\left( g_{\tau_L} * 1_{[0,1]} \right)(x) = 0$ for $x < 0$, we finally obtain

$$f_{E \to I}(t_0 + k) = \sum_{j=-\infty}^{k} f_{S \to E}(t_0 + j) \int_{k-j}^{k-j+1} \left( g_{\tau_L} * 1_{[0,1]} \right)(x) \, dx \tag{7}$$

However, this equation includes infinitely many unknown parameters ($f_{S \to E}(t_0 + j)$, where $j$ is an integer smaller or equal to $k$), making parameter estimation challenging. To overcome this, we approximated the equation (7) by assuming individuals were exposed at a constant rate, $r$, before time $t_0$ (Fig 2b (iii)):

$$f_{S \to E}(t_0 + j) \approx r \quad for \quad j < 0 \tag{8}$$

Here, we found that this constant rate $r$ is closely related to the $E(t_0 - 1)$. Specifically, we can express $E(t_0 - 1)$ as a function of $r$:

$$E(t_0 - 1) = \int_0^\infty \widetilde{f}_{S \to E}(t_0 - 1 - x) \int_x^\infty g_{\tau_L}(y) dy dx \approx r \cdot \int_0^\infty \int_x^\infty g_{\tau_L}(y) dy dx \tag{9}$$

This equation arises from the fact that the people exposed at time $t_0 - 1 - x$ can remain in the exposed (E) compartment at time $t_0 - 1$ only if their latent period ($\tau_L$) is greater than $x$. By applying Fubini's theorem to this equation, we can further simplify the equation as follows:

$$E(t_0 - 1) \approx r \cdot \int_0^\infty \int_x^\infty g_{\tau_L}(y) dy dx = r \cdot \int_0^\infty \int_0^y g_{\tau_L}(y) dx dy = r \cdot \int_0^\infty y g_{\tau_L}(y) dy = r \cdot \mathbb{E}[\tau_L] \tag{10}$$

where $\mathbb{E}[\tau_L]$ is the mean of the latent period. This equation suggests that the constant rate $r$ is proportional to the $E(t_0 - 1)$:

$$r \approx \frac{E(t_0 - 1)}{\mathbb{E}[\tau_L]} \tag{11}$$

Plugging in equations (8) and (11) to equation (7), we can derive the final equation:

$$f_{E \to I}(t_0 + k) \approx \frac{E(t_0 - 1)}{\mathbb{E}[\tau_L]} \int_{k+1}^\infty \left(g_{\tau_L} * 1_{[0,1]}\right)(s) ds + \sum_{j=0}^k f_{S \to E}(t_0 + j) \int_{k-j}^{k-j+1} \left(g_{\tau_L} * 1_{[0,1]}\right)(s) ds \tag{12}$$

This final equation holds for every given data $(f_{E \to I}(t_0), \ldots, f_{E \to I}(t_0 + n))$, and this system of $n + 1$ equations can be written in a matrix form (Fig 2c):

$$\begin{pmatrix} f_{E \to I}(t_0) \\ \vdots \\ f_{E \to I}(t_0 + n) \end{pmatrix} = \frac{E(t_0 - 1)}{\mathbb{E}[\tau_L]} \begin{pmatrix} Q_1 \\ \vdots \\ Q_{n+1} \end{pmatrix} + \begin{pmatrix} P_0 & \cdots & 0 \\ \vdots & \ddots & \vdots \\ P_n & \cdots & P_0 \end{pmatrix} \begin{pmatrix} f_{S \to E}(t_0) \\ \vdots \\ f_{S \to E}(t_0 + n) \end{pmatrix} \tag{13}$$

where $Q_{k+1} = \int_{k+1}^\infty \left(g_{\tau_L} * 1_{[0,1]}\right)(s) ds$ and $P_{k-j} = \int_{k-j}^{k-j+1} \left(g_{\tau_L} * 1_{[0,1]}\right)(s) ds$. From this matrix equation, we established the data loss function which is minimal when the left and right sides of the equation (13) are similar.

$$data\ loss = \sum_{k=0}^n \left(f_{E \to I}(t_0 + k) - \frac{E(t_0 - 1)}{\mathbb{E}[\tau_L]} Q_{k+1} - \sum_{j=0}^k f_{S \to E}(t_0 + j) P_{k-j}\right)^2 \tag{14}$$

We aimed to find the unknown parameters $(E(t_0 - 1), f_{S \to E}(t_0), \ldots, f_{S \to E}(t_0 + n))$ by minimizing the data loss. However, as the number of unknown parameters ($n + 2$), exceeds the number of equations ($n + 1$), the parameters cannot be determined solely from the data loss. This leads us to incorporate the additional regularization loss for the $f_{S \to E}$ parameters. For this regularization, we employed the second-order derivative of daily incidence of exposure, because typically one day is insufficient to make a drastic increase or decrease in daily change of the exposed population. As a result, we derived the final total loss function.

$$\begin{aligned}
total\ loss &= \mathcal{L}(E(t_0 - 1), f_{S \to E}(t_0), \ldots, f_{S \to E}(t_0 + n)) = data\ loss + regularization\ loss \\
&= \sum_{k=0}^n \left(f_{E \to I}(t_0 + k) - \frac{E(t_0 - 1)}{\mathbb{E}[\tau_L]} Q_{k+1} - \sum_{j=0}^k f_{S \to E}(t_0 + j) P_{k-j}\right)^2 \\
&+ \sum_{k=1}^{n-1} \left(f_{S \to E}(t_0 + k + 1) - f_{S \to E}(t_0 + k) - (f_{S \to E}(t_0 + k) - f_{S \to E}(t_0 + k - 1))\right)^2
\end{aligned} \tag{15}$$

Finally, we calculated the initial condition of E ($E(t_0)$) from the estimated parameters ($E(t_0 - 1)$, $f_{S \to E}(t_0)$) and available information ($f_{E \to I}(t_0)$) by using following formula:

$$E(t_0) = E(t_0 - 1) - f_{E \to I}(t_0) + f_{S \to E}(t_0)$$

## Parameter estimation from the loss function

To find the value of $E(t_0 - 1)$, $f_{S \to E}(t_0)$, ..., $f_{S \to E}(t_0 + n)$ minimizing the total loss function (i.e., equation (12)), we utilized the Limited-memory Broyden-Fletcher-Goldfarb-Shanno (L-BFGS) method. This algorithm is a gradient-based quasi-Newton approach designed for solving large-scale optimization problems. The optimization process incorporates boundary conditions to ensure biological plausibility. These constraints guarantee that the estimated values remain non-negative and do not exceed the maximum population size S, preserving the physical meaning of the parameters.

## Construction and simulation of the SEIR model with Gamma-distributed latent and infectious periods

We compared the accuracy of Hist-I and Hist-D by using simulation data from the SEIR model mimicking the latent period of COVID-19. For this, we constructed the SEIR model following the previous study by Hong and Eom et al. [19]:

$$\frac{dS(t)}{dt} = -\beta \frac{S(t)I(t)}{N}$$

$$\frac{dE(t)}{dt} = \beta \frac{S(t)I(t)}{N} - Flow_{E \to I}(t)$$

$$\frac{dI(t)}{dt} = Flow_{E \to I}(t) - Flow_{I \to R}(t) \tag{16}$$

$$\frac{dR(t)}{dt} = Flow_{I \to R}(t)$$

where $\beta$ is the transmission rate and $N$ is the number of the entire population (i.e., $N = S(0) + E(0) + I(0) + R(0)$). $Flow_{E \to I}(t)$ and $Flow_{I \to R}(t)$ indicates the history-dependent rate of transition from compartment E to I and I to R, respectively. These rates were calculated as follows:

$$Flow_{E \to I}(t) = \int_0^t \beta \frac{S(u)I(u)}{N} g_1(t - u)du + E(0) \widetilde{g_1}(t)$$

$$Flow_{I \to R}(t) = \int_0^t \beta \frac{S(u)I(u)}{N}(g_1 * g_2)(t - u)du + E(0)(\widetilde{g_1} * g_2)(t) + I(0)\widetilde{g_2}(t)$$

$$\widetilde{g_i}(t) = \frac{1}{\mathbb{E}[g_i]} \int_t^\infty g_i(v)dv, \quad i = 1, 2 \tag{17}$$

where $g_1(t)$ and $g_2(t)$ represent the probability density functions of the distribution of the latent period and infectious period, respectively, and $\widetilde{g_1}(t)$ and $\widetilde{g_2}(t)$ represent the probability density functions of sojourn times of individuals initially in compartments E and I, respectively [19].

We set initial conditions as $S(0) = 9558153$, $E(0) = 0$, $I(0) = 1$, $R(0) = 0$ to mimic the initial stage of COVID-19 in Seoul, where the whole population of Seoul is susceptible except for one infectious individual. Additionally, we assumed that the latent period follows the Gamma distribution with shape 4.06 and scale 1.35, based on a previous study that fitted a gamma distribution to observed data on the COVID-19 latent period [24]. The infectious period was assumed to follow the Gamma distribution with shape 30 and scale 0.2, as estimated in a previous study using same COVID-19 data employed in this study [19]. Simulating this model using Heun's method, we obtained daily numbers of $S(t)$, $E(t)$, $I(t)$, and $R(t)$, which were then converted to the daily incidence of exposure ($f_{S \to E}$) data and daily incidence of becoming infectious ($f_{E \to I}$) data as follows:

$$f_{S \to E}(t) = -\Delta S(t) = -(S(t) - S(t-1))$$

$$f_{E \to I}(t) = -\Delta E(t) + f_{S \to E}(t) = -(E(t) - E(t-1)) - (S(t) - S(t-1)) \tag{18}$$

These daily exposed and infectious people datasets were employed to compare the performances between Hist-I and Hist-D (Fig 3c-3e). Then, to further consider the possible measurement errors in the real-world, we applied multiplicative noise ($e^{U(-\sigma, \sigma)}$) to the given data (i.e., daily incidence of becoming infectious, $f_{E \to I}$) (Fig 3f) and compared the estimation accuracy of Hist-I and Hist-D (Fig 3g). Additionally, we incrementally increased the noise level ($\sigma$) by 0.1 to assess whether Hist-D maintained its superiority under higher noise conditions (Fig 3h). To ensure the reliability and stability of our findings, this process was iterated 10 times, and their average RMSE and MAPE were reported (Fig 3h). Lastly, we simulated a sudden shift in epidemic dynamics by adjusting $\beta$ during the simulation (Fig 3i), changing its initial value of 0.4 by multiplying it by factors of 1/3, 1/2, 1, 2, 3 at $t = 140$.

## Data collection and preprocessing

For the real-world data analysis, we utilized contact tracing data of COVID-19 in Seoul, Republic of Korea from January 20th, 2020 to November 25th, 2020. From this data, the period from August 13th, 2020 to November 25th, 2020 was chosen as the testing set for Hist-D and Hist-I as it includes both the increasing phase and decreasing phase of exposed individuals. The dataset contains individual contact dates, symptom onset dates, and confirmation dates of COVID-19 cases. While confirmation dates were complete, only 35% and 63% of contact dates and symptom onset dates were available, respectively, leading us to use 21% of the total data that had complete information on contact dates, symptom onset dates, and confirmation dates.

From these data, we extracted the number of daily incidence of exposure ($f_{S \to E}$) and daily incidence of becoming infectious ($f_{E \to I}$), by assuming that individuals become "exposed" at their contact dates and "infectious" at their symptom onset dates. Then, we calculated the population in the E compartment by setting $E(t_0) = 0$ on January 20th, 2020, the date of the first officially confirmed COVID-19 case in Seoul [19], and cumulatively summing the difference of $f_{S \to E}$ and $f_{E \to I}$. The resulting values were used as true values of $E(t_0)$. Additionally, the distribution of the incubation period was obtained empirically, by calculating the time difference between the date of symptom onset and the contact date for each case: the probability of an incubation period of 5 days is the ratio of cases whose difference between symptom onset date and contact date is 5 days.

These data are protected and are not available due to data privacy laws. The Korea Public Institutional Review Board Designated by Ministry of Health and Welfare waived the need for ethical approval for the collection and analysis of the real-world data since the data was anonymized and none of the individuals were identifiable (reference number: P01-202404-01-016).

## Uncertainty quantification using the Markov chain Monte Carlo (MCMC) method

We quantified the parametric and prediction uncertainties of Hist-D using Bayesian inference. We set the likelihood of given observed daily incidence of becoming infectious, $f_{E \to I}$, as follows:

$$f_{E \to I}(t_0 + k) \sim \text{Poisson}\left( \frac{E(t_0-1)}{\mathbb{E}[\tau_L]} Q_{k+1} + \sum_{j=0}^{k} f_{S \to E}(t_0 + j) P_{k-j} \right) \tag{19}$$

where $P_{k-j}$ and $Q_{k+1}$ are known constants introduced in equation (10). The prior distributions of parameters were assumed as follows:

$$E(t_0 - 1) \sim \text{LogNormal}(\hat{E}(t_0 - 1), \sigma_E^2)$$

$$f_{S \to E}(t_0 + j) \sim \text{LogNormal}\left(\hat{f}_{S \to E}(t_0 + j), k\right) \quad j = 0, \ldots, n$$

$$k \sim \text{LogNormal}\left(\mu_k, \sigma_k^2\right) \tag{20}$$

where $\text{LogNormal}(x, y)$ denotes the lognormal distribution with mean $x$ and variance $y$ and $\hat{E}(t_0 - 1), \hat{f}_{S \to E}(t_0), \ldots, \hat{f}_{S \to E}(t_0 + n)$ denotes the point estimates of $E(t_0 - 1), f_{S \to E}(t_0), \ldots, f_{S \to E}(t_0 + n)$, obtained by utilizing Hist-D. We used $\sigma_E = \sigma_k = 100$, and $\mu_k = 20$.

We used an MCMC method to sample the parameters $(E(t_0 - 1), f_{S \to E}(t_0), \ldots, f_{S \to E}(t_0 + n))$ from their posterior distribution defined by the likelihood in [19] and the priors in [20]. To be more specific, we performed 100,000 iterations of sampling $E(t_0 - 1), f_{S \to E}(t_0), \ldots, f_{S \to E}(t_0 + n)$ from their posterior distributions, by using a Hamiltonian Monte Carlo (HMC) algorithm with the No-U-Turn Sampler (NUTS). We generated the posterior samples of $E(t_0)$ by cumulatively summing up the difference between posterior sample of daily incidence of exposure and given data: $E(t_0) = E(t_0 - 1) + f_{S \to E}(t_0) - f_{E \to I}(t_0)$. Then, the credible interval for Hist-D is calculated by determining the range between 0.025 and 0.975 quantiles of the posterior samples of $E(t_0)$.

## Quantification and statistical analysis

In this study, functions to estimate the initial condition and simulate all scenarios were developed by the authors in the programming languages R (version 4.3.2) and Stan (version 2.32.2) and Rstan (version 2.32.6).

## Supporting information

**S1 File. Hist-D.** Computational package for Hist-D.
(ZIP)

**S1 Text. Computational package for Hist-D.**
(DOCX)

**S2 Text. Estimation of the reproduction number heavily depends on the initial conditions.**
(DOCX)

**S3 Text. Hist-D is more accurate than Hist-I, even after Hist-I was modified.**
(DOCX)

**S4 Text. Hist-D enhances the accuracy of reproduction number estimation.**
(DOCX)

**S1 Fig. Estimation of reproduction number is heavily dependent on the initial condition of E.** The plot displaying the fold error between the estimated reproduction number ($\hat{\Re}$) and the true reproduction number ($\Re$) across various bias levels in $E(0)$. $E(0)$ was initially $E_0 = 50$, and it changed to the $E_1$. When $\Re = 2$ (red curve), $E_1/E_0 = 2^{2.5}$ led to a 37.9%

relative error and $E_1/E_0 = 2^{-2.5}$ led to a 19.4% relative error. When $\Re = 4$ (blue curve), $E_1/E_0 = 2^{2.5}$ led to a 42.5% relative error and $E_1/E_0 = 2^{-2.5}$ led to a 34.1% relative error. (b) The plot showing the estimated time-varying reproduction number (($\hat{\Re}(t)$)) under varying levels of bias in the initial condition of $E$: $\frac{E(0)}{4}$, $\frac{E(0)}{2}$, $E(0)$, $2E(0)$, and $4E(0)$, where $E(0)$ is the original initial condition value. The influence of bias in $E(0)$ continues noticeably well after the time point where the initial condition was estimated, and decreases over time, leading to convergence in the estimated reproduction numbers. (EPS)

**S2 Fig. The superiority of Hist-D was preserved even after the modification of Hist-I as done in Rauch et al. (a) The graph comparing the true $E(t_0)$ (light gray-colored bars) and the estimated $E(t_0)$ ($\hat{E}(t_0)$).** The modified Hist-I underestimates the $E(t_0)$ (b) The scatter plot displaying the error between estimated $E(t_0)$ ($\hat{E}(t_0)$) and true $E(t_0)$ across different levels of true $E(t_0)$. Estimation from Hist-D (green squares) shows smaller errors compared to the modified Hist-I (red triangles). (c) The bar plot showing the root mean squared error (RMSE; bars) and mean absolute percentage error (MAPE; line) of the modified Hist-I and Hist-D. RMSE and MAPE were reduced by 80% and 53%, respectively, when Hist-D were utilized, compared to the modified Hist-I. (EPS)

**S3 Fig. Hist-D outperforms both Hist-I and the modified Hist-I even in the early phase of the epidemic dynamics.** (a) The graph comparing the true $E(t_0)$ (light gray-colored bars) and the estimated $E(t_0)$ ($\hat{E}(t_0)$) for time $t_0 = 10 - 80$. (b) The scatter plot displaying the error between estimated $E(t_0)$ ($\hat{E}(t_0)$) and true $E(t_0)$ across different levels of true $E(t_0)$. Estimation from Hist-D (green squares) shows smaller errors compared to both Hist-I (red triangles) and the modified Hist-I (blue circles). (c) The bar plot showing the root mean squared error (RMSE; bars) and mean absolute percentage error (MAPE; line) of Hist-I, the modified Hist-I, and Hist-D. Hist-D reduced both RMSE and MAPE by 81% compared to Hist-I, whereas the modified Hist-I reduced both by 47%. (EPS)

**S4 Fig. Hist-D enhances the accuracy of reproduction number estimation.** Boxplots of the posterior samples of the reproduction number ($\hat{\Re}$) obtained from IONISE with initial conditions estimated by Hist-D (green) and Hist-I (red). IONISE combined with Hist-D accurately estimated the reproduction number, while using Hist-I introduced considerable bias. Here, the posterior samples were normalized by the true value ($\Re$) employed for generating the simulation data depicted in Fig 3c. (EPS)

**S1 Table. Hist-D is more accurate than Hist-I under various parameter conditions.** The table shows the reduction in RMSE (former) and MAPE (latter) achieved by Hist-D relative to Hist-I under the same conditions as Fig 3c–3e, but with varied latent and infectious period parameters. (DOCX)

## Author contributions

**Conceptualization:** Sunhwa Choi, Jae Kyoung Kim.

**Data curation:** Boseung Choi.

**Formal analysis:** Dongju Lim, Kyeong Tae Ko, Hyukpyo Hong.

**Funding acquisition:** Hyojung Lee, Boseung Choi, Won Chang, Sunhwa Choi, Jae Kyoung Kim.

**Investigation:** Dongju Lim, Kyeong Tae Ko, Hyukpyo Hong, Hyojung Lee, Boseung Choi, Won Chang, Sunhwa Choi, Jae Kyoung Kim.

**Methodology:** Dongju Lim, Kyeong Tae Ko, Jae Kyoung Kim.

**Project administration:** Sunhwa Choi, Jae Kyoung Kim.

**Supervision:** Jae Kyoung Kim.

**Validation:** Dongju Lim, Kyeong Tae Ko, Sunhwa Choi, Jae Kyoung Kim.

**Visualization:** Dongju Lim, Kyeong Tae Ko.

**Writing – original draft:** Dongju Lim, Kyeong Tae Ko, Sunhwa Choi, Jae Kyoung Kim.

**Writing – review & editing:** Dongju Lim, Kyeong Tae Ko, Hyukpyo Hong, Hyojung Lee, Boseung Choi, Won Chang, Sunhwa Choi, Jae Kyoung Kim.

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
