## [Decision Letter · Decision Letter 0]

5 May 2025

PCOMPBIOL-D-25-00204

A History-Dependent Approach for Accurate Initial Condition Estimation in Epidemic Models

PLOS Computational Biology

Dear Dr. Kim,

Thank you for submitting your manuscript to PLOS Computational Biology. After careful consideration, we feel that it has merit but does not fully meet PLOS Computational Biology's publication criteria as it currently stands. Therefore, we invite you to submit a revised version of the manuscript that addresses the points raised during the review process.

Please submit your revised manuscript within 30 days Jul 05 2025 11:59PM. If you will need more time than this to complete your revisions, please reply to this message or contact the journal office at ploscompbiol@plos.org. Please include the following items when submitting your revised manuscript:

We look forward to receiving your revised manuscript.

Kind regards,

Nik J. Cunniffe

Academic Editor

PLOS Computational Biology

Roger Kouyos

Section Editor

PLOS Computational Biology

**Additional Editor Comments :**

Thank you for sending this very nice paper to PLOS Computational Biology. I was fortunate to receive comments from three reviewers, all of whom have engaged with the paper and provided useful and relevant comments. All three agree with my own view that this is an interesting and potentially important contribution, considering how the initial number of latently infected individuals can be better estimated from data, going beyond a broad brush assumption based on the initial infected population and the expected latent period. However, in different ways all reviewers comment on how the work could be slightly better motivated and/or generalised: I was struck by R3's comment about transmission before symptoms, which is common for pathogens of a range of host taxa, and this should at least be discussed. I was also struck by R1's comments around making the case more strongly that the so-called Hist-I assumption is, in fact, "conventional"/widely used, and agree that no consideration of the method of stages is surprising. While these issues can, probably, be handled by adding more text, some of the other more detailed issues raised by R3 in their review might require additional computational work/reworking of some figures. I will also be very interested to read the authors' response to R3's comment around the potential for error introduced by assuming a constant rate of exposure for an epidemic that is growing exponentially, and it might be that some further tests with synthetic data might be helpful here. Both R1 and R3 raise minor issues with the supplied code or its documentation, and for this type of article I think these are important to address in full. Nevertheless, this appears at this stage to be a very nice article, and I look forward to seeing a revision, which at this stage I intend to send to (at least) R1 again (perhaps also R2 and R3, depending on what is said in the response to reviewers letter).

**Journal Requirements:**

At this stage, the following Authors/Authors require contributions: Dongju Lim, Kyeong Tae Ko, Hyojung Lee, Boseung Choi, Won Chang, Sunhwa Choi, and Jae Kyoung Kim. Please ensure that the full contributions of each author are acknowledged in the "Add/Edit/Remove Authors" section of our submission form.

3) We have noticed that you have uploaded Supporting Information files, but you have not included a complete list of legends. Please add a full list of legends for your Supporting Information files (HIstD.zip) after the references list.

4) In the online submission form you indicate that your data is not available for proprietary reasons and have provided a contact point for accessing this data. Please note that your current contact point is a co-author on this manuscript. According to our Data Policy, the contact point must not be an author on the manuscript and must be an institutional contact, ideally not an individual. Please revise your data statement to a non-author institutional point of contact, such as a data access or ethics committee, and send this to us via return email. Please also include contact information for the third party organization, and please include the full citation of where the data can be found.

**Reviewers' comments:**

Reviewer's Responses to Questions

**Comments to the Authors:**

**Please note that two reviews are uploaded as attachments.**

Reviewer #1: Please see attachment.

Reviewer #2: The review is uploaded as an attachment.

Reviewer #3: In this article, the authors address the inference of size of the latent exposed class in SEIR-like epidemic models. They contrast the bias in estimates of the size of the latent class when making the (common) assumption that the size of the initial latent class is determined by the initial size of the infected population and the expected latent period, with their preferred set of assumptions: that allows a distribution of time since infected in the exposed class, assuming that prior to the initial time point the risk of exposure (moving from S to E) is constant. The authors construct a loss function with regularisation (penalising rapid changes in the rate of exposure) and generate posterior estimates for $E(t_0)$ using hamiltonian MCMC implemented in Stan, arguing that this approach is computationally efficient and results in improved estimates of the epidemic size (including latent exposures). They use both synthetic and detailed case data -- that includes information on observed E(t) -- for SARS-CoV-2 in South Korea to validate these results, and explore the impact of noise of different magnitude on their ability to recover the size of the exposed class.

This paper is a commendable attempt to consider the consequences of an oft used approximation for epidemic initial conditions. The detailed epidemiological case data, including exposure and onset dates, used to validate their approach is a significant strength. I have some suggestions that may help clarify the approach and the consequences of the results.

* The adopted model structure can accommodate general distributions for the latent and infectious periods, but given for SARS-CoV-2 and other pathogens transmission prior to symptom onset is common, and capturing the generation time appropriately is important for accurate estimates of the initial and time-varying reproduction numbers, can you briefly clarify (perhaps repeating information on Hong et al. [28]?) how $g_1$ and $g_2$ are chosen to ensure an appropriate generation time in this model?

* I presume the main focus of the inference is E(t) and I(t), and can see that $g_{1,2}$ are fixed, but it would be good to be clear(er) about whether particular $\beta$ (and its evolution in fig 3i), are being inferred, and if appropriate showing some of the bivariate posterior distributions.

* Figure 1 of the supplementary material demonstrates the bias introduced to estimates of $R_0$ generated by bias in E(t). Apologies if I have missed this in the discussion, but It would be nice to understand how this bias evolves in estimates of $R(t)$ in this modelling framework - i.e. if/when do these biases in IC become irrelevant for predicting the current model state?

* When discussing the limitations of the model, is it also possible that the discrepancy in coverage of CIs also due to assumption of constant exposure prior to $t_0$? E.g. Often epidemics may begin with 'super-spreading' events that the model isn't capturing (e.g. https://www.nature.com/articles/s41467-023-42612-9)

* Consider adding the version of Stan used to run the code, there was a syntax error in the data block for the current version (though easy to fix).

* The inference did not run for me when using the input data file 'SimulData_cde.csv' due an incorrect date format. It did run as for the file specified in the code, though with many divergent transitions. Please consider checking the instructions for running the code, and reporting on the convergence diagnostics for your MCMC chains.

**Have the authors made all data and (if applicable) computational code underlying the findings in their manuscript fully available?**

Reviewer #1: Yes

Reviewer #2: Yes

Reviewer #3: Yes

PLOS authors have the option to publish the peer review history of their article (what does this mean? ). If published, this will include your full peer review and any attached files.

**Do you want your identity to be public for this peer review?** For information about this choice, including consent withdrawal, please see our Privacy Policy .

Reviewer #1: No

Reviewer #2: No

Reviewer #3: No

**Figure resubmission:**
---

## [Decision Letter · Decision Letter 1]

5 Aug 2025

PCOMPBIOL-D-25-00204R1

A History-Dependent Approach for Accurate Initial Condition Estimation in Epidemic Models

PLOS Computational Biology

Dear Dr. Kim,

Thank you for submitting your manuscript to PLOS Computational Biology. After careful consideration, we feel that it has merit but does not fully meet PLOS Computational Biology's publication criteria as it currently stands. Therefore, we invite you to submit a revised version of the manuscript that addresses the points raised during the review process.

Please submit your revised manuscript within 30 days Oct 05 2025 11:59PM. If you will need more time than this to complete your revisions, please reply to this message or contact the journal office at ploscompbiol@plos.org. Please include the following items when submitting your revised manuscript:

We look forward to receiving your revised manuscript.

Kind regards,

Roger Dimitri Kouyos

Section Editor

PLOS Computational Biology

**Additional Editor Comments :**

Thank you for engaging so positively with the reviewers' comments. As you will see, the reviewers who had substantive comments are now almost entirely satisfied, as am I. I will leave it up to you to decide whether - or not - to address the remaining comments from Reviewer #1; despite what is said below about "only permit corrections to spelling, formatting or significant scientific errors" you should be able to address these - if you wish - as part of the process of moving to publication.

**Journal Requirements:**

1) Please ensure that the funders and grant numbers match between the Financial Disclosure field and the Funding Information tab in your submission form. Note that the funders must be provided in the same order in both places as well. Currently, the order of the grants is different in both places especially  "NIMS-B24810000" and "SSTF-BA1902-01."

**Reviewers' comments:**

Reviewer's Responses to Questions

Reviewer #1: I thank the authors for their detailed responses to my comments and those of the other reviewers, and am mostly satisfied with the responses. I have just a few remaining minor comments:

1. In the abstract, please modify “this history-independent method yields serious bias in the estimation of the initial condition” to “this history-independent method can yield serious bias in the estimation of the initial condition when the latent period is not exponentially distributed”. Similarly, in the author summary, change “this unrealistic assumption leads to serious errors” to “this unrealistic assumption can lead to serious errors”.

2. RE my initial comment 4, I agree that the method of stages is (at least slightly) less flexible than the approach here, although it’s not quite true that the number of stages must be treated as a tuning parameter – if a parametric or empirical latent period distribution is available (as is assumed in this study), the number of stages and transition rate(s) between them could be obtained by fitting the model’s (implicit) Erlang-distributed latent period distribution to best match the “true” distribution. Please revise the relevant part of the discussion accordingly.

3. I found this sentence in the Discussion a little weird: “First, our methods are derived from ODE-based infectious disease models, though PDE-based models are also used to account for spatial effects in disease spread”, since PDE models aren’t particularly common in the (applied) modelling literature. It would be nice instead to discuss the extension of this approach to other types of models more generally (for example, stochastic compartmental models and network-based models).

Reviewer #2: The authors have done a nice job with the revision and have addressed all of my concerns.

Reviewer #3: Thank you for thoroughly responding to my comments, I am happy to recommend acceptance.

**Have the authors made all data and (if applicable) computational code underlying the findings in their manuscript fully available?**

Reviewer #1: Yes

Reviewer #2: Yes

Reviewer #3: Yes

PLOS authors have the option to publish the peer review history of their article (what does this mean? ). If published, this will include your full peer review and any attached files.

**Do you want your identity to be public for this peer review?** For information about this choice, including consent withdrawal, please see our Privacy Policy .

Reviewer #1: No

Reviewer #2: No

Reviewer #3: No

**Figure resubmission:**
---

## [Editor Report · Decision Letter 2]

14 Aug 2025

Dear Professor Kim,

We are pleased to inform you that your manuscript 'A History-Dependent Approach for Accurate Initial Condition Estimation in Epidemic Models' has been provisionally accepted for publication in PLOS Computational Biology.

Best regards,

Nik J. Cunniffe

Academic Editor

PLOS Computational Biology

Roger Kouyos

Section Editor

PLOS Computational Biology

---

## [Editor Report · Acceptance letter]

PCOMPBIOL-D-25-00204R2

A History-Dependent Approach for Accurate Initial Condition Estimation in Epidemic Models

Dear Dr Kim,

I am pleased to inform you that your manuscript has been formally accepted for publication in PLOS Computational Biology. Your manuscript is now with our production department and you will be notified of the publication date in due course.

With kind regards,

Narmatha Raju, M.Sc
